# Genetic differentiation of *Oncomelania hupensis robertsoni* in hilly regions of China: Using the complete mitochondrial genome

Jing Song[1,2,3☉], Hongqiong Wang[1,2☉], Shizhu Li[4], Peijun Qian[4], Wenya Wang[4], Meifen Shen[1,2], Zongya Zhang[1,2], Jihua Zhou[1,2], Chunying Li[5], Zaogai Yang[5], Yuwan Hao[5]*, Chunhong Du[1,2]*, Yi Dong[1,2]*

1 Department of Schistosomiasis Control and Prevention, Yunnan Institute of Endemic Disease Control and Prevention, Dali, People's Republic of China, 2 Yunnan Key Laboratory of Natural Focus Disease Control Technology, Dali, People's Republic of China, 3 Yunnan Provincial Key Laboratory of Public Health and Biosafety, Kunming, People's Republic of China, 4 National Key Laboratory of Intelligent Tracking and Forecasting for Infectious Diseases, National Institute of Parasitic Diseases at Chinese Center for Disease Control and Prevention, Chinese Center for Tropical Diseases Research, NHC Key Laboratory of Parasite and Vector Biology, WHO Collaborating Center for Tropical Diseases, National Center for International Research on Tropical Diseases, Shanghai, People's Republic of China, 5 School of Public Health, Kunming Medical University, Kunming, People's Republic of China

☉ These authors contributed equally to this work.
* haoyw@nipd.chinacdc.cn (YH); dch6890728@163.com (CD); dydali@sina.com (YD)

**Data Availability Statement:** Detailed information about the organization of the mitochondrial genome is uploaded as S2 File. The sequences are

## Abstract

### Objective

*Oncomelania hupensis robertsoni* is the only intermediate host of *Schistosoma japonicum* in hilly regions of south-west China, which plays a key role during the transmission of Schistosomiasis. This study aimed to sequence the complete mitochondrial genome of *Oncomelania hupensis robertsoni* and analyze genetic differentiation of *Oncomelania hupensis robertsoni*.

### Methods

Samples were from 13 villages in Yunnan Province of China, with 30 *Oncomelania hupensis* snails per village, and the complete mitochondrial genome was sequenced. A comprehensive analysis of the genetic differentiation of *Oncomelania hupensis robertsoni* was conducted by constructing phylogenetic trees, calculating genetic distances, and analyzing identity.

### Results

A total of 26 complete mitochondrial sequences were determined. The length of genome ranged from 15,181 to 15,187 bp, and the base composition of the genome was A+T (67.5%) and G+C content (32.5%). This genome encoded 37 genes, including 13 protein-coding genes, 2 rRNA genes, and 22 tRNA genes. The phylogenetic trees and identity analysis confirmed that *Oncomelania hupensis robertsoni* was subdivided into *Oncomelania hupensis robertsoni* Yunnan strain and Sichuan strain, with a genetic distance of 0.0834.

available in NCBI (https://www.ncbi.nlm.nih.gov/), with accession number: OR661779, OR661780, OR661781, OR661782, OR661783, OR661784, OR661785, OR661786, OR661787, OR661788, OR661789, OR661790, OR661791, OR661792, OR661793, OR661794, OR661795, OR661796, OR661797, OR661798, OR661799, OR661800, OR661801, OR661802, OR661803, OR661804.

**Funding:** This work was supported by National Key Research and Development Program of China (No. 2021YFC2300800, 2021YFC2300803 to SZL), Open project of Key Laboratory of Parasite and Vector Biology of National Health Commission (NHCKFKT2023-08 to SZL), Dali Prefecture Science and Technology Plan Project (D2022ZA0115 to JS, 20242901A020013 to HQW), Open Project of Yunnan Provincial Key Laboratory of Public Health and Biosafety (KLPHB-2023-04 to JS). The funders had no role in study design, data collection and analysis, decision to publish, or preparation of the manuscript.

**Competing interests:** The authors declare that no competing interests exist.

*Oncomelania hupensis robertsoni* Yunnan strain was further subdivided into two sub-branches, corresponding to "Yunnan North" and "Yunnan South", with a genetic distance of 0.0216, and the samples exhibited over 97% identity.

## Conclusion

*Oncomelania hupensis robertsoni* is subdivided into *Oncomelania hupensis robertsoni* Yunnan strain and Sichuan strain. *Oncomelania hupensis robertsoni* Yunnan strain exhibits a higher level of genetic identity and clear north-south differentiation. This work reported the first mitochondrial genome of *Oncomelania hupensis robertsoni* Yunnan strain, which could be used as an important reference genome for *Oncomelania hupensis*, and also provide the important information for explaining the distribution pattern of *Oncomelania hupensis robertsoni* and control of *Schistosoma japonicum*.

### Author summary

*Oncomelania hupensis robertsoni* is the only intermediate host of *Schistosoma japonicum* in hilly regions of south-west China, its distribution area directly determines the epidemic range of schistosomiasis. This study sequenced 26 complete mitochondrial genomes of *Oncomelania hupensis robertsoni* and analyzed their genetic differentiation. *Oncomelania hupensis robertsoni* is subdivided into *Oncomelania hupensis robertsoni* Yunnan strain and Sichuan strain. *Oncomelania hupensis robertsoni* Yunnan strain exhibits a higher level of genetic identity and clear north-south differentiation. To the best of our knowledge, this work reported the first mitochondrial genome of *Oncomelania hupensis robertsoni* Yunnan strain, which could be used as an important reference genome for *Oncomelania hupensis*, and also provide the important information for explaining the distribution pattern of *Oncomelania hupensis robertsoni* and control of *Schistosoma japonicum*.

## Introduction

Schistosomiasis is a neglected tropical parasitic disease that has imposed severe burden worldwide affecting almost 240 million people, and more than 700 million people live in endemic areas [1]. Six species of schistosomes are known to cause human infection: *Schistosoma japonicum* (*S. japonicum*), *S. mansoni*, *S. haematobium*, *S. malayensis*, *S. intercalatum* and *S. mekongi* [2]. Among these species, *S. japonicum* is deemed the most virulent due to its ability to produce a greater number of eggs compared to other species, resulting in severe disease pathology [3,4]. Notably, China is one of the major endemic countries of *S. japonicum*, with current measures primarily focused on the monitoring and control of *Oncomelania hupensis* (*O. hupensis*) snail, the sole intermediate host of *S. japonicum* [5].

In China, *O. hupensis* snails primarily inhabit the 12 southern provinces in the middle and lower reaches of the Yangtze River, its geographical distribution range is extensive, with notable climate variations and complex array of environmental types [6]. Significant genetic differentiation and variation occur in the *O. hupensis* due to the influence of geographical isolation [7], encompassing four subspecies: *Oncomelania hupensis hupensis* (*O. h. hupensis*), *Oncomelania hupensis robertsoni* (*O. h. robertsoni*), *Oncomelania hupensis tangi* (*O. h. tangi*), and *Oncomelania hupensis guangxiensis* (*O. h. guangxiensis*) [8]. *O. hupensis* residing in different

geographical regions display morphological differences and genetic variations, along with varying susceptibility to *S. japonicum* [8–10]. Considering the close genetic interaction between *S. japonicum* and its intermediate host, the *O. hupensis*, in terms of co-evolution [11,12], understanding the genetic differentiation and classification of *O. hupensis* is of great significance for understanding the transmission of schistosomiasis and concerning snail control through focal molluscicides [13].

Yunnan Province was previously a severe area for schistosomiasis endemism in hilly regions of China and still harbors a substantial population of snails, i.e., *O. h. robertsoni* [8]. Population genetic experiments suggest that Yunnan Province may be the original location for *O. hupensis* in Chinese mainland [14,15]. After initially entering Yunnan Province from the Himalayan Mountains, *O. hupensis* subsequently migrated to the Sichuan Plain, which is connected to the Yangtze River, and then dispersed further to the east coast of Chinese mainland. Therefore, studying the genetic differentiation of *O. h. robertsoni* is important for understanding the origins of *O. hupensis* in the Chinese mainland. Currently, *O. hupensis* is predominantly found in Yunnan Province particularly in areas such as ditches, grassland, field ridges, wasteland, and dry land. Its distribution is relatively isolated and patchy, with substantial fragmentation [16]. High mountain or watershed barriers exist between some distribution areas, creating geographically separated and non-contiguous regions. Given this unique geographical environment, the genetic diversity of *O. h. robertsoni* in Yunan Province could exhibit a certain degree of heterogeneity, posing challenges for the development of targeted snail monitoring and controlling.

In recent years, mitochondrial DNA (mtDNA) has found widespread application as a molecular marker in numerous biosystematics studies [17–20]. However, current research on *O. hupensis* mtDNA mostly focuses on individual gene fragments such as cytochrome c oxidase1 (COX1), cytochrome b (CYTB), ribosomal 12S and 16S RNA sequence [6,21–28], the results obtained have certain limitations to reveal the *O. hupensis* of population structure and genetic variation [29]. Fewer studies obtained the complete mitochondrial sequence of *O. hupensis* [23,30], and lacking the sequence of *O. h. robertsoni* in Yunan Province, this merits more research attention.

The study herein aimed to sequence the complete mitochondrial genome of the *O. h. robertsoni*, and to explore the differentiation characteristics of this population. Our results may provide the molecular biology-based theoretical foundation for understanding the genetic differentiation and population structure of *O. hupensis* and the genetic information for monitoring and control of *O. h. robertsoni* in hilly region of China.

## Material and methods

### Source of *O. h. robertsoni*

In the existing snail distribution area of Yunnan Province of China, 13 villages were selected as snail collection sites after considering various factors such as snail density, geographical location, altitude, water system, and environmental type (**Fig 1** and **Table 1**).

### Extraction of total DNA from *O. hupensis*

Dead *O. hupensis* snails were selected and removed using the crawling method. Briefly, the snails were placed in the center of a petri dish with a diameter of 15cm, after which a few drops of water were poured over the snails. A mesh cover was placed over the petri dish to prevent the live snails from crawling out. The set up was placed at room temperature (25°C) over 24 hours, after which the ones that crawled from the center were regarded as live [31,32]. Thirty live snails were used for DNA extraction.

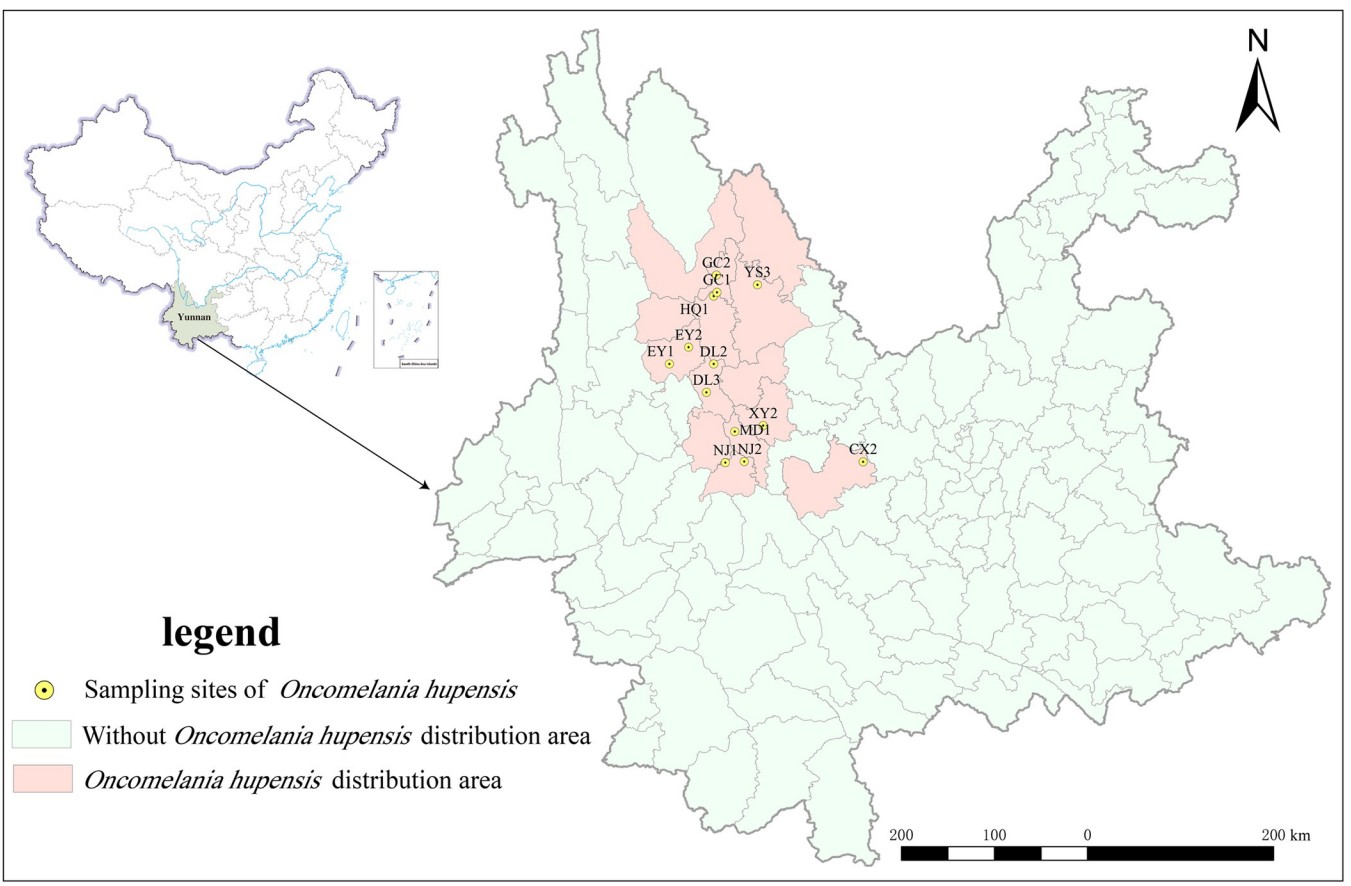

**Fig 1. Distribution of sampling sites of *O. h. robertsoni* in Yunnan Province of China.** The base layer of the map is from the publicly accessible GADM dataset (https://gadm.org/download_country.html).

DNA was extracted from 7mm head-foot muscle of individual snails using the Qiagen DNeasy Blood & Tissue Kit [Paisennuo Biotechnology (Shanghai) Co., Ltd].

### Primer design, PCR amplification and sequencing

The amplification primers are presented in **Table 2**. The entire mitogenome was divided into 16 overlapping fragments according to the sequence length.

### Sequence assembly and genomic annotation

All sequence fragments were filtered to remove those that were not normally reported, such as PCR "amplification failure" and sequencing "bimodal mutation" (two major peaks appear in the sequencing results). The LaserGene 7.1.0 (DNAStar, Madison, Wisconsin, USA) was used to contig assemble the 16 sequences of each sample, remove the sequences that could not be assembled into closed circular forms. The overlapping regions of adjacent sequences that could be assembled into closed circular forms were adjusted, and the entire sequence was manually checked for base identification based on peak shape.

The assembled and edited sample sequences were submitted to National Center for Biotechnology Information (NCBI, https://www.ncbi.nlm.nih.gov/) for sequence alignment and direction confirmation. Based on the gene order of the most similar mitochondrial sequence (the less divergent mitochondrial genome sequence to all sample sequences is the strain of *O.*

**Table 1. Location of *O. h. robertsoni* collection.**

| Collection site (Code) | Habitat environment | No. samples | Collection date | Latitude | Longitude | Altitude (m) |
|---|---|---|---|---|---|---|
| Leqiu village (NJ1) | Grassland | 98 | 20-Dec-2021 | 100.3470° E | 25.0412° N | 1716.90 |
| Anding village (NJ2) | Grassland | 102 | 21-Dec-2021 | 100.5298° E | 25.0510° N | 1250.10 |
| Caizhuang village (MD1) | Vegetable field | 100 | 27-Dec-2021 | 100.4384° E | 25.3431° N | 1666.48 |
| Xiaoqiao village (XY2) | Ditch | 100 | 28-Dec-2021 | 100.7120° E | 25.3988° N | 1915.78 |
| Qiandian village (EY1) | Grassland | 99 | 10-Jan-2022 | 99.8189° E | 25.9810° N | 1907.34 |
| Yongle village (EY2) | Ditch | 100 | 11-Jan-2022 | 99.9900° E | 26.1565° N | 2042.25 |
| Wuxing village (DL2) | Ditch | 98 | 7-Jan-2022 | 100.2334° E | 25.9937° N | 1794.73 |
| Xiaocen village (DL3) | Ditch | 101 | 11-Jan-2022 | 100.1643° E | 25.7204° N | 1944.66 |
| Lianyi village (HQ1) | Dry land | 94 | 23-Dec-2021 | 100.2334° E | 26.6516° N | 2177.05 |
| Yangwu Village (YS3) | Ditch | 100 | 29-Dec-2021 | 100.6571° E | 26.7613° N | 1558.40 |
| Sanyi Village (GC1) | Pond | 109 | 24-Dec-2021 | 100.2664° E | 26.6883° N | 2258.92 |
| Dongyuan Village (GC2) | Wasteland | 90 | 24-Dec-2021 | 100.2596° E | 26.8549° N | 2336.20 |
| Cangling Village (CX2) | Wasteland | 92 | 14-Jan-2022 | 101.6790° E | 25.0505° N | 1781.32 |

The selected villages were Leqiu (NJ1) and Anding (NJ2) in Nanjian County, Caizhuang (MD1) in Midu County, Xiaoqiao (XY2) in Xiangyun County, Qiandian (EY1) and Yongle (EY2) in Eryuan County, Wuxing (DL2) and Xiaocen (DL3) in Dali County, Lianyi (HQ1) in Heqing County of Dali City; Yangwu Village (YS3) in Yongsheng County, Sanyi Village (GC1) and Dongyuan Village (GC2) in Gucheng District of Lijiang City; and Cangling Village (CX2) in Chuxiong County of Chuxiong City. The habitat conditions of snails at each collection site are presented in **Figs A-N in S1 File**, with approximately 100 snails were collected from each sampling site.

*h. robertsoni* in Sichuan Province [33], accession number: JF284691), the sample sequences were adjusted and ordered to have the same starting gene sequence as the COX1 gene in the SCXC strain of *O. hupensis* mitochondrion. The edited sequences and all complete mitochondrial sequences of *O. hupensis* retrieved from NCBI were aligned, and sequence composition analysis was performed using MEGA11.0.9 [34] (https://www.megasoftware.net/). Finally, the gene structure annotation was performed using MITOS [35] (http://mitos.bioinf.uni-leipzig.de) by Beijing Tsingke Biotech Co., Ltd.

## Construction of phylogenetic tree

The full length of mitochondrial genome sequences obtained were used for constructing phylogenetic trees. A total of 14 complete mitochondrial genome sequences of *O. hupensis* strain were retrieved from the NCBI and used as the reference sequences.

The reference sequences included those from the Philippines (accession number: JF284698.1 FLB), Sichuan Province (accession number: JF284697 SCMS, JF284691 SCXC), Fujian Province (accession number: JF284695 FJFQ), Zhejiang Province (accession number: JF284694 ZJJH), Guangdong Province (accession number: MN200239 Guangdong), Jiangsu Province (accession number: JF284688 JSYZ), Anhui Province (accession number: JF284686 AHGD-1, JF284687 AHGD-2), Hunan Province (accession number: JF284692 HNYY), Hubei Province (accession number: JF284689 HBGA, JF284690 HBJL), Jiangxi Province (accession number: JF284693 JXSR), and Guangxi Province (accession number: JF284696 GXBS).

Phylogenetic trees were constructed using MEGA11.0.9 [34] by the four methods of Maximum Parsimony (MP) (parameter settings: MP Search Method: SPR, No. of Initial Tree:10, MP Search level:1, Max No. of Tree to Rain:100), Maximum Likelihood (ML) (parameter settings: Model/Method: Tamura-Nei model, ML Heuristic Method: NNI, Initial Tree for ML: Make initial tree automatically, Substitutions to Include: GTR model), Minimum Evolution (ME) (parameter settings: Model/Method: Maximum Composite Likelihood, Substitutions to

**Table 2. Amplification primers for *O. hupensis* mitochondrial genome.**

| Primers Code | Primer Sequences | Length | Site | PCR fragment size (bp) |
|---|---|---|---|---|
| MT.F1 | AACAAATCATAAAGATATTGGGAC | 24 | 18 | 912 |
| MT.R1 | GCAATAATTATCGTAGCCGC | 20 | 929 | |
| MT.F2 | TCAGCTAAGAAAGAAACGTTTG | 22 | 775 | 1546 |
| MT.R2 | CAATTGAGGCATTAAAGAATACT | 23 | 2320 | |
| MT.F3 | CAATCATTCATTTATACCAATTGT | 24 | 2518 | 1374 |
| MT.R3 | ACAAGCAGTGTTTAGGGCAC | 20 | 3531 | |
| MT.F4 | CATTTGTTGGGGAGAATTAAC | 21 | 3264 | 1305 |
| MT.R4 | CTTTTCAGCGAGAGCGAC | 18 | 4568 | |
| MT.F5 | CGTCAAATCAAGGTACAGCC | 20 | 4390 | 1238 |
| MT.R5 | GCTCGATAGGGTCTTCTTGTC | 21 | 5627 | |
| MT.F6 | TACTCTGACCGTGCGAAGG | 19 | 5448 | 777 |
| MT.R6 | GCGACTGCTAATAAAATACAGAT | 23 | 6224 | |
| MT.F7 | GTGAGCCAGGTCAGTTTCT | 19 | 5913 | 1277 |
| MT.R7 | AATATAATTACTGTCATTCAGGAC | 24 | 7189 | |
| MT.F8 | TGACATGAAAAAGATTTTTACC | 22 | 7028 | 1184 |
| MT.R8 | CGAGTTAATGTTGCATTATCA | 21 | 8211 | |
| MT.F9 | TCTTATTGTGGTAGTAAAAATTTG | 24 | 7612 | 1289 |
| MT.R9 | GTTAAAACTAAATAAAATGTTCTCAT | 26 | 8982 | |
| MT.F10 | CTTGAAAATAAACTAAGAGTGCTC | 24 | 8844 | 887 |
| MT.R10 | ATATAATTGTTAATAGTAGT | 20 | 9730 | |
| MT.F11 | CCAACTCTCCTTATTATTTTAGG | 23 | 9616 | 1074 |
| MT.R11 | AACATGGTTGGGTAAAATTAAA | 22 | 10689 | |
| MT.F12 | CTCCTTCACTGAATTCCTGC | 20 | 10552 | 744 |
| MT.R12 | GGTAGCCAGGCTGAAAAT | 18 | 11295 | |
| MT.F13 | ACTTACAATTATAGCAAATCGAAT | 24 | 11110 | 1328 |
| MT.R13 | ATCTTCAGTGCCATGCTCTA | 20 | 12437 | |
| MT.F14-1 | CAATCTCAGCTCAGCCGCTA | 20 | 12181 | 1554 |
| MT.R14-1 | TTGCCCCCAAAAATGTTCTT | 20 | 13734 | |
| MT.F14-2 | TTGCCCCCAAAAATGTTCTT | 25 | 12354 | 1439 |
| MT.R14-2 | CGTTTAGACAGCACTCACCC | 20 | 13792 | |
| MT.F15 | TTCCTAAATGACTGAGAATAAGTG | 24 | 13325 | 1150 |
| MT.R15 | TGAAACGGAAAAATTCCAG | 19 | 14473 | |
| MT.F16 | TTGTCCTTTTTCTTATGTTTTCA | 23 | 14119 | 1250 |
| MT.R16 | CAAATGCATGTGCTGTAACA | 20 | 180 | |

The reaction mixture contained 2.5 units DreamTaq Green DNA Polymerase, 2 * polymerase chain reaction (PCR) buffer, 2.5 mmol/L dNTPs, 10 μmol/L primers, and >30ng genomic DNA in a final volume of 25 μL. Thermal cycle programmed for 180 seconds at 95˚C as initial denaturation, followed by 35 cycles of 10 sec at 98˚C for denaturation, 5 sec at 58˚C as annealing, 60 sec at 72˚C for extension, and final extension at 72˚C for 5 min. The primer synthesis, PCR amplification, PCR product purification and sequencing were performed by Shanghai Xianghong Biotech Co., Ltd.

Include: d: Transitions + Transversions), and Neighbor-Joining (NJ) (parameter settings: Model/Method: Maximum Composite Likelihood, Substitutions to Include: d: Transitions + Transversions), respectively, and each method had repeats of 1000 bootstrap value.

## Sequence genetic distance and identity analysis

Genetic distances within and between groups were calculated based on alignment results using the MEGA11.0.9 [34]. Sequence identity was calculated using the LaserGene 7.1.0 (DNAStar,

Madison, Wisconsin, USA). The average nucleotide similarity among 40 *O. hupensis* sequences was calculated using Fastani 1.32 (https://github.com/ParBLiSS/FastANI) [36], identity analysis was performed by comparing genome sequence similarities combined with Blast, and the Average nucleotide identity (ANI) heat map was generated by Tbtools 2.001 [37] (https://github.com/CJ-Chen/TBtools-II/releases).

## Results

### Mitochondrial genome structure of *O. h. robertsoni*

A total of 26 complete mitochondrial genome sequences from 13 sampling sites were obtained.

The length of genome ranged from 15,181bp to 15,187bp, with an average length of 15,185bp. The average contents of A, T, C and G bases were 29.7%, 37.8%, 15.6% and 16.9%, respectively, in which the A+T content (67.5%) was higher than the G+C (32.5%) content. This genome encoded 37 genes, including 13 protein-coding genes, 2 rRNA genes, and 22 tRNA genes. Detailed information about the organization of the mitochondrial genome is presented in **Tables A-I in S2 File**.

For example, in NJ1-01 sequence, there were 19 intergenic regions totaling 272bp, ranging in length from 1bp to 68bp, with the largest intergenic region located between ND5 and trnF (gaa) (68bp), and 5 gene overlap regions totaling 17bp (**Fig 2**).

### Phylogenetic tree construction

Phylogenetic tree is presented in **Fig 3**. The topology of the phylogenetic tree constructed by the four methods of MP, ML, ME, and NJ was consistent (Fig 3), And the bootstrap values of all branches were greater than 70, with most exceeding 90, indicating a high degree of reliability [38].

The genotype from the Philippines was used as the outgroup, and the other sequences clustered into 2 major branches, *O. h. robertsoni*, and the other including genotypes from *O. h. tangi* and *O. h. hupensis*. *O. h. robertsoni* was subdivided into *Oncomelania hupensis robertsoni* Yunnan strain (*O. h. r.* Yunan strain) and Sichuan strain.

In addition, *O. h. r.* Yunan strain was further subdivided into two sub-branches, tentatively named "Yunnan North Branch" and "Yunan South Branch". As shown in **Fig 4,** the North branch included samples distributed in the Jinsha River Watershed (i.e. HQ1, YS3, GC1, GC2), located in northern of Yunnan Province. The South branch included samples distributed in the Lancang River Watershed (i.e., EY1, EY2, and DL3), Yuan River Watershed (i.e., WS1, MD1, XY2, NJ1 and NJ2), Longchuan River Watershed (i.e., CX2), and Jinsha River Watershed (i.e. DL2), located in western, central and southern regions of Yunnan Province. There are obvious geographic barriers between the Jinsha River Watershed and the Lancang River, Yuan River, and Longchuan River Watershed, and these barriers include the high mountains and canyons formed by the Hengduan and Yunling Mountain ranges.

### Genetic distance and identity analysis

The average genetic distance between the *O. h. robertsoni* and *O. h. hupensis* was 0.113. The average genetic distance between the *O. h. hupensis* (Zhejiang, Guangdong, Jiangsu, Anhui, Hunan, Hubei, Jiangxi Provinces) was 0.0216, and the average genetic distance between the *O. h. r.* Yunan strain and *O. h. r.* Sichuan strain was 0.0834 in *O. h. robertsoni*. Moreover, the average genetic distance within *O. h. r.* Yunan strain was 0.0129, while the genetic distance between the Northern Branch and Southern Branch was 0.0216.

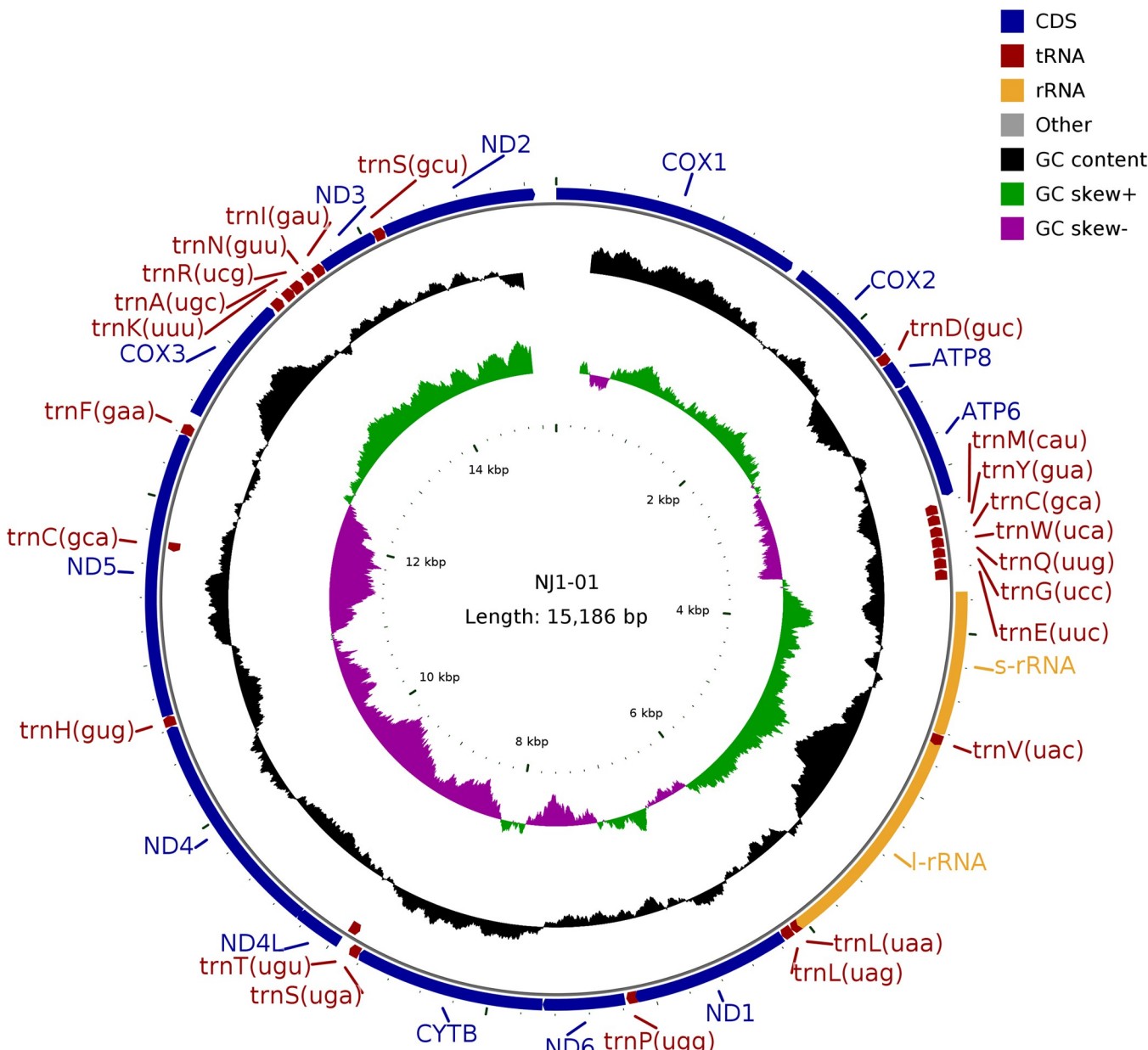

**Fig 2. Representative circular map of the mitochondrial genome of *O. h. robertsoni*.** The colored squares distributed inside and outside the circle represent different mitochondrial genes, gene taxa of the same function are represented using the same color.

Some clustering information in the phylogenetic tree can be visualized in the ANI heat map, as well as the degree of identity among the samples. As shown in **Fig 5**, the identity among the samples of *O. h. r.* Yunnan strain was above 97%, forming an orange rectangle, which contained two small red rectangles, tentatively named "North Yunnan Branch" and "South Yunnan Branch". Except for the *O. h. robertsoni* and *O. h. quadrasi*, other subspecies of *O. hupensis* forme a rectangle in the lower right corner.

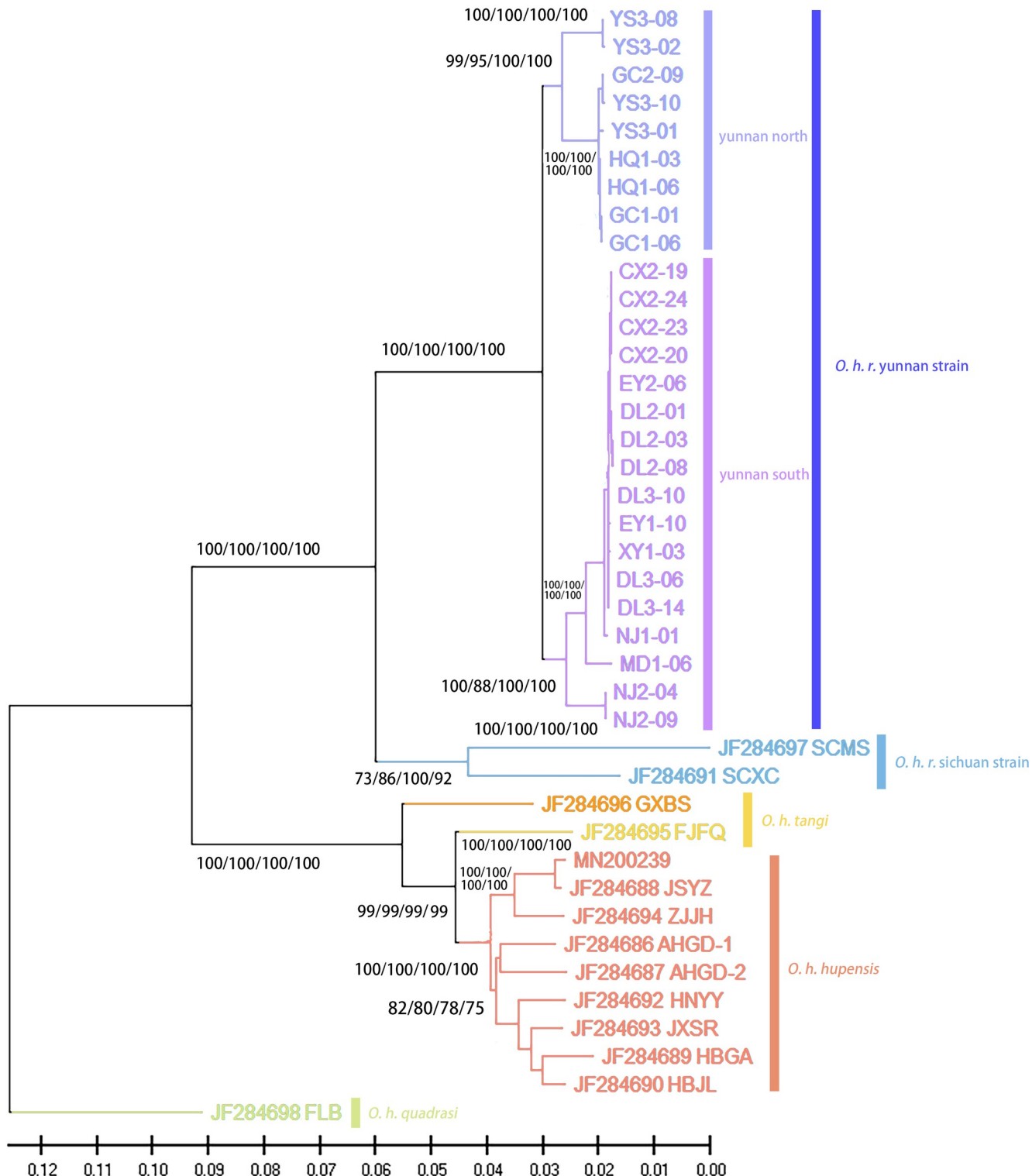

**Fig 3. Phylogenetic tree constructed by Maximum Likelihood method.** Values on nodes represent bootstrap support percentage for ML/MP/ME/NJ, Scale bar represent one nucleotide substitution for every 100 nucleotides. *O. h. quadrasi* (Philippines genotype) was used to the tree as an outgroup.

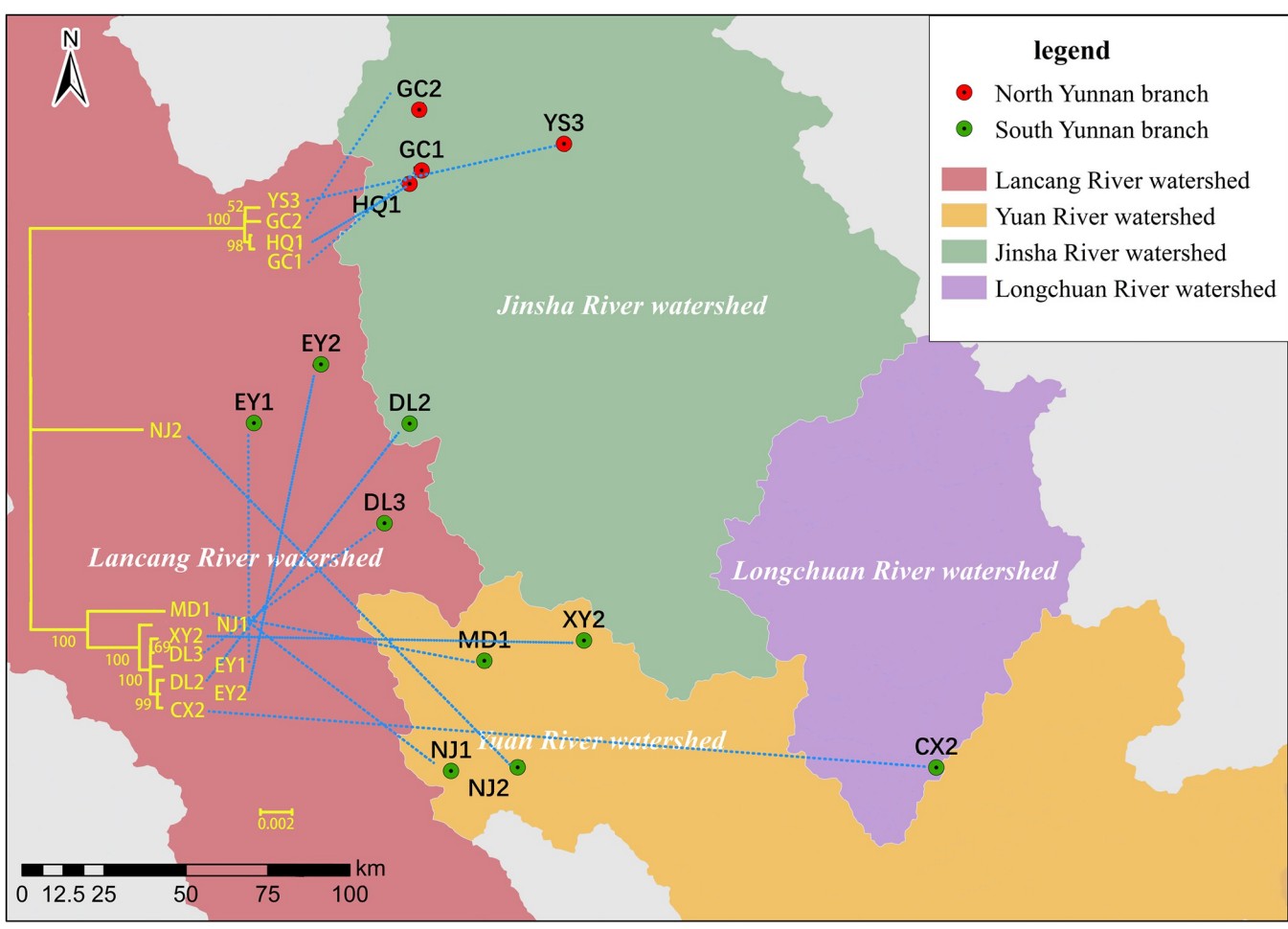

**Fig 4. Geographical locations of North-South Yunnan branch of *O. h. robertsoni* (Phylogenetic tree constructed by ML method).** The base layer of the map from the publicly accessible Resource and Environmental Science Data Center (https://www.resdc.cn/DOI/DOI.aspx?DOIID=44).

## Discussion

As the risk of schistosomiasis transmission or outbreaks may recur in regions with snails [39], monitoring and control of *O. hupensis* snail is the important ways to interrupt the transmission of disease. Given the emerging, re-emerging, and persistent habitats of snails [40,41], understanding their genetic differentiation might be useful for their successful monitoring and control. In this study, we collected 13 representative *O. h. robertsoni* snail populations from hilly endemic schistosomiasis areas in Yunnan Province of China, and obtained 26 complete mitochondrial sequences for the analysis of genetic differentiation. Our results may provide the important information for explaining the distribution pattern of *O. h. robertsoni* and control of *S. japonicum* in such regions.

In recent years, genetic sequence analysis has gained widespread utilization in the fields of phylogenetics and population genetics [42–45]. MtDNA has many advantages over nuclear genes for phylogenetic inference and classification due to its easy amplification with a large number of available conserved primers, lack of recombination, introns, non-coding sequences, and maternal inheritance, which simplify the complexity of phylogenetic studies [42,46–49]. Furthermore, mtDNA evolves rapidly, exhibits high variability within populations, and has high sensitivity for resolving closely related species [50]. Currently, mitochondrial gene

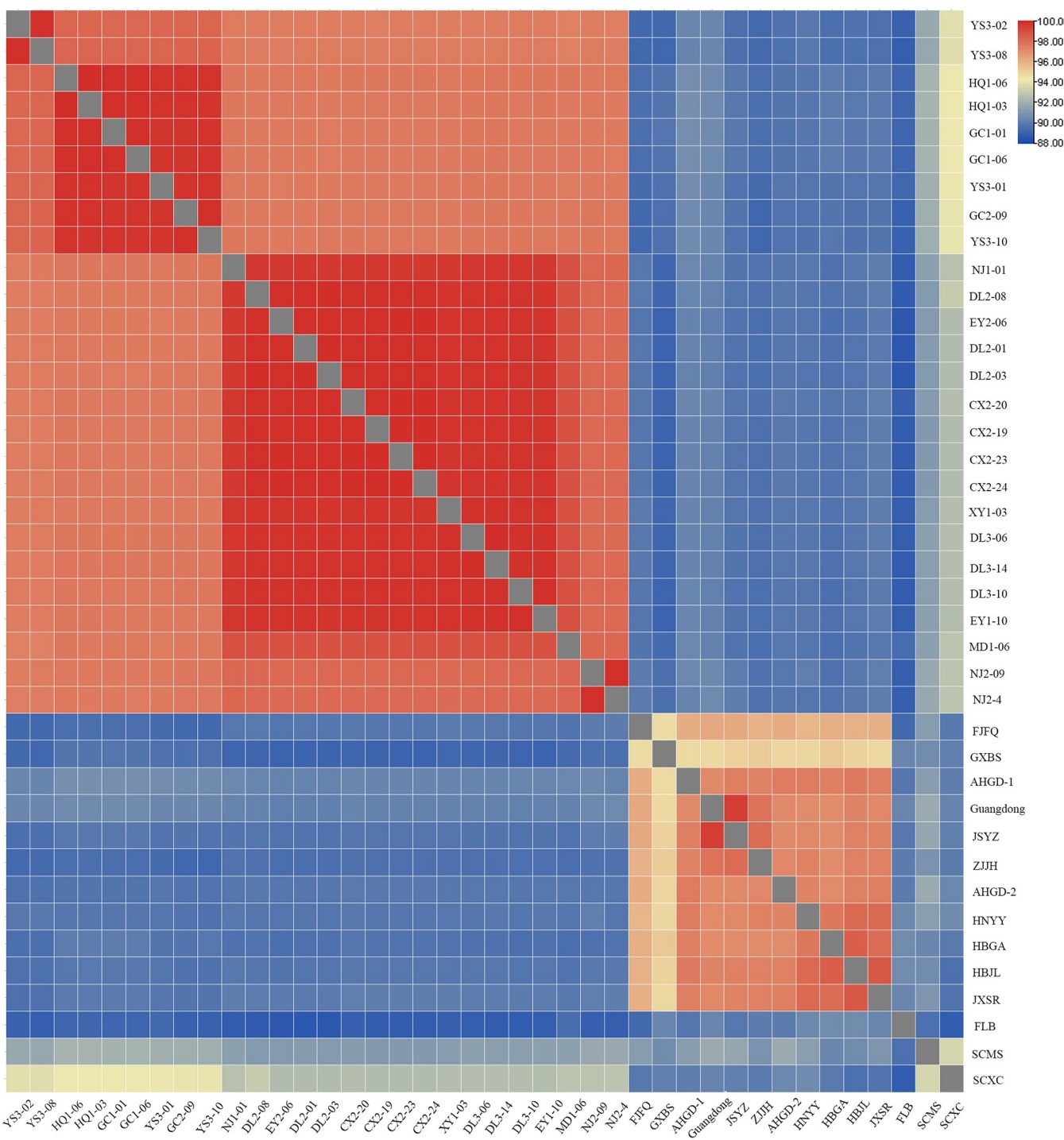

**Fig 5. Average nucleotide identity heat map of mitochondrial sequence of *O. hupensis*.**

fragments or mitochondrial genomes are the common markers for analyzing genetic polymorphism and genetic variation of *O. hupensis*. However, individual gene fragments may no longer provide the necessary level of identification and comprehensive phylogenetic analysis [51,52]. While previous studies have reported the mitochondrial genome sequence of *O.*

*hupensis* and its phylogenetic analysis, they have not conducted a systematic analysis of *O. h. r.* Yunnan strain, and the number of samples involved in the previous research on the mitochondrial genome was also limited [23]. To our knowledge, this work reported the first mitochondrial genome of *O. h. r.* Yunnan strain, which could be used as an important reference genome for *O. hupensis*, and the gene composition and distribution closely mirrored that of the mitochondrial genome of *O. h. hupensis* [23]. Additionally, genetic data of *O. h. hupensis* can also be used for targeted development of snail control drugs. Different populations of snails may differ in their susceptibility to certain drugs at the genetic level, and genetic data from snails can be used to optimize drug formulations and methods of use [14].

Based on the phylogenetic trees [38], the clustering of *O. h. r.* Yunnan strain and Sichuan strain into one large branch, supporting the previous classification of the *O. h. robertsoni* [8,14,53]. In addition to the genetic evidence, the morphology of *O. h. robertsoni* differs from other *O. hupensis* snails. Compared with other subspecies, the morphological characteristics of *O. h. robertsoni* are smooth shell surface without longitudinal ribs, no labial ridges, and uniform growth of each conch layer [25].

In addition, *O. h. r.* Yunnan strain and *O. h. r.* Sichuan strain formed two distinct branches, showing clear differentiation. It's worth noting that the genetic distance between Yunnan and Sichuan populations was significantly greater than that observed between other subspecies of *O. hupensis*, but the genetic distance between other subspecies of *O. hupensis* was narrower, reflecting higher identity and less pronounced differentiation. Our results were consistent with other previous findings based COX1 gene and 16S ribosomal RNA gene, the existence of at least two major phylogroups within *O. h. robertsoni* was demonstrated [27,54,55]. The genetic differences between *O. hupensis* populations are associated with different environmental adaptations for survival and susceptibility to parasite infection [56]. The fact that snail populations in some regions are genetically more different from other regions implies that snails in that region have unique adaptations for survival or risk of schistosomiasis transmission. Such as, decreased genetic differentiation of *Biomphalaria pfeifferi* snail populations due to Diama dam construction in the Senegal River Basin, associated with the major local outbreak of *S. mansoni* in 1990 [57,58]. These findings may provide important information for formulating targeted measures for schistosomiasis control.

MtDNA is an organelle gene, and its DNA content is relatively small compared with that of nuclear genes, which provides limited genetic information. Phylogenetic tree analysis showed that *O. h. r.* Yunnan strain is subdivided into two subgroups, tentatively named "Yunnan North Branch" and " Yunnan South Branch". The result of phylogenetic tree of *O. h. r.* Yunnan strain was consistent with the other study using microsatellite markers [59], suggesting that the mitochondrial genome can be used reliably in the genetic evolutionary analysis of the *O. hupensis*. Additionally, when considering the ANI heat map and genetic distance, the average genetic distance within the *O. h. r.* Yunnan strain was 0.0129, indicating a high degree of identity and limited differentiation. Although the *O. h. r.* Yunnan strain is divided into two small subgroups, genetic differentiation between these subgroups is not significant, as evidenced by a genetic distance of 0.0216, which is similar to the average genetic distance observed between provinces of *O. h hupensis* in this study. *O. h. r.* Yunnan strain is mainly distributed in three river watersheds: the Jinsha River, Lancang River, and Yuan River. Their distribution pattern is fragmented by high mountain barriers or isolation, resulting in three distinct regions: western Yunnan, central Yunnan, and southern Yunnan. Even within the same county, the distribution areas may be characterized by small or point-like "islands" due to mountainous terrain or river barriers. Previous research has explored the presence of these "island" characteristics in *O. h. r.* Yunnan strain [16], but according to the data in this study, there is not a clear "island" feature. Instead, these seems to be a correlation with the distribution pattern of the

river watershed. Notably, the *O. hupensis* in the "North Yunnan" subgroup exclusively occupies the Jinsha River watershed, while the remaining sampling points are distributed across the other three river watershed, with the exception of Wuxing Village in Dali City, located on the southern edge of the Jinsha River Watershed. Although three river watersheds are separated by obvious geographical barriers, samples from each watershed cluster together in the same branch of the phylogenetic tree, indicating a high degree of identity and no apparent differentiation based on the distribution of river watershed. The divergence between the northern and southern lineages of Yunnan snails may result from either unique natural factor in the Jinsha River Watershed such as different soil salinity level, PH, and wetness, or geographic isolation due to the significant distance separating the two lineages. Further research is required to determine the exact cause of this divergence. In addition, Wuxing Village in Dali City, situated within the Jinsha River Watershed, grouped with the "South Yunnan" lineage on the phylogenetic tree. This deviation may be due to its position on the edge of the Jinsha River Watershed, making it more susceptible to gene flow with other snail populations in Dali that are geographically closer.

Furthermore, the genetic distance between the two geographical strains in Sichuan Province was 0.0704, surpassing the average genetic distance of 0.0291 observed among the other subspecies of *O. hupensis*, with the exception of *O. h. robertsoni* and the Philippine *O. hupensis*. This suggested a lower degree of identity and more pronounced differentiation. However, only two geographical strains of *O. hupensis* from Sichuan province were included in this study, and further research with larger sample sizes is needed to verify the actual differentiation status of *O. hupensis* in Sichuan Province.

Major strengths of this study lie in sequence assembly to avoid the computational challenges and possible errors of computational assembly of short reads, we adjusted the sequencing of overlapping sequence regions and manually checked the entire sequence to ensure a high level of accuracy in our results [60]. However, there is limitation to this study. Due to extensive schistosomiasis control efforts, the snails in Gejiu city, located in the southern part of Yunnan Province, have been eliminated [61]. Consequently, during *O. hupensis* sampling for this study, we were unable to collect *O. hupensis* from this region. Gejiu city was relatively isolated within the previous endemic regions of schistosomiasis in Yunnan Province, we hypothesis that *O. hupensis* in this area may be a separate population. The genetic differentiation analysis of *O. hupensis* in this area was not included, which may introduce some limitations to our findings.

## Conclusion

In summary, our study successfully obtained 26 complete mitochondrial genome sequence of *O. h. robertsoni* and conducted a comprehensive analysis of its genetic differentiation. Our results indicate that *O. h. robertsoni* is subdivided into *O. h. r* Yunnan strain and Sichuan strain, *O. h. r* Yunnan strain exhibits a higher level of genetic identity and clear north-south differentiation. These findings provide the important information for explaining the distribution pattern of *O. h. robertsoni* and might also have implications for the development of more effective strategies for the control of schistosomiasis in hilly regions.

Additionally, 26 complete mitochondrial genome sequence of *O. h. robertsoni* may be as reference sequence for analysis of genetic differentiation and classification of *O. hupensis*.

## Supporting information

**S1 File. Habitat characteristics of *O. h. robertsoni* in sampling site for each environmental type. Fig A.** Habitat environment of *O. hupensis* in sampling site of NJ1. **Fig B.** Habitat

environment of *O. hupensis* in sampling site of NJ2. **Fig C.** Habitat environment of *O. hupensis* in sampling site of MD1. **Fig D.** Habitat environment of *O. hupensis* in sampling site of XY2. **Fig E.** Habitat environment of *O. hupensis* in sampling site of EY1. **Fig F.** Habitat environment of *O. hupensis* in sampling site of EY2. **Fig G.** Habitat environment of *O. hupensis* in sampling site of DL2. **Fig H.** Habitat environment of *O. hupensis* in sampling site of DL3. **Fig I.** Habitat environment of *O. hupensis* in sampling site of WS1. **Fig J.** Habitat environment of *O. hupensis* in sampling site of HQ1. **Fig K.** Habitat environment of *O. hupensis* in sampling site of YS3. **Fig L.** Habitat environment of *O. hupensis* in sampling site of GC1. **Fig M.** Habitat environment of *O. hupensis* in sampling site of GC2. **Fig N.** Habitat environment of *O. hupensis* in sampling site of CX2.
(DOCX)

**S2 File. Detailed information about the organization of 26 mitochondrial genome of *O. h. robertsoni*. Table A**. The organization of the mitochondrial genome of *O. h. r.* Yunnan strain in sampling sites of Nanjian County. **Table B.** The organization of the mitochondrial genome of *O. h. r.* Yunnan strain in sampling sites of Midu County (MD1-06). **Table C.** The organization of the mitochondrial genome of *O. h. r.* Yunnan strain in sampling sites of Xiangyun County (XY2-03). **Table D.** The organization of the mitochondrial genome of *O. h. r.* Yunnan strain in sampling sites of Eryuan County. **Table E**. The organization of the mitochondrial genome of *O. h. r.* Yunnan strain in sampling sites of Dali City. **Table F.** The organization of the mitochondrial genome of *O. h. r.* Yunnan strain in sampling sites of Heqing County. **Table G.** The organization of the mitochondrial genome of *O. h. r.* Yunnan strain in sampling sites of Yongsheng County. **Table H.** The organization of the mitochondrial genome of *O. h. r.* Yunnan strain in sampling sites of Lijiang City. **Table I.** The organization of the mitochondrial genome of *O. h. r.* Yunnan strain in sampling sites of Chuxiong City.
(DOCX)

## Acknowledgments

We thank Schistosomiasis control institutions in Yunnan Province for their valuable helps in *O. hupensis* collection.

## Author Contributions

**Conceptualization:** Jing Song, Hongqiong Wang, Yuwan Hao, Chunhong Du, Yi Dong.

**Data curation:** Jing Song, Hongqiong Wang, Peijun Qian, Wenya Wang.

**Formal analysis:** Jing Song, Hongqiong Wang.

**Funding acquisition:** Jing Song, Hongqiong Wang, Shizhu Li.

**Investigation:** Meifen Shen, Zongya Zhang, Jihua Zhou, Chunying Li, Zaogai Yang, Chunhong Du.

**Methodology:** Jing Song, Hongqiong Wang.

**Project administration:** Yi Dong.

**Software:** Jing Song.

**Supervision:** Shizhu Li, Yuwan Hao, Chunhong Du, Yi Dong.

**Validation:** Hongqiong Wang, Zongya Zhang.

**Visualization:** Yi Dong.

**Writing – original draft:** Jing Song, Hongqiong Wang.

**Writing – review & editing:** Shizhu Li, Yuwan Hao, Chunhong Du, Yi Dong.

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
