## [Decision Letter · Decision Letter 0]

9 Jun 2024

Dear MD Dong,

Thank you very much for submitting your manuscript " Genetic differentiation of Oncomelania hupensisrobertsoni in hilly regions of China : using the complete mitochondrial genome" for consideration at PLOS Neglected Tropical Diseases. As with all papers reviewed by the journal, your manuscript was reviewed by members of the editorial board and by several independent reviewers. In light of the reviews (below this email), we would like to invite the resubmission of a significantly-revised version that takes into account the reviewers' comments. 

We cannot make any decision about publication until we have seen the revised manuscript and your response to the reviewers' comments. Your revised manuscript is also likely to be sent to reviewers for further evaluation.

Sincerely,

Luc E. Coffeng, MD PhD

Academic Editor

Audrey Lenhart

Section Editor

Reviewer's Responses to Questions

**Key Review Criteria Required for Acceptance?**

**Methods**

-Are the objectives of the study clearly articulated with a clear testable hypothesis stated?

-Is the study design appropriate to address the stated objectives?

-Is the population clearly described and appropriate for the hypothesis being tested?

-Is the sample size sufficient to ensure adequate power to address the hypothesis being tested?

-Were correct statistical analysis used to support conclusions?

-Are there concerns about ethical or regulatory requirements being met?

Reviewer #1: The objective of the study, which is to sequence the complete mitochondrial genome of Oncomelania hupensis robertsoni and analyze the genetic differentiation of Oncomelania hupensis robertsoni, is clearly stated. The study design is appropriate in addressing the objective. The sample size and source population are likewise adequate.

I have also listed below specific comments regarding the Methods section:

Line Comment

144 This statement needs further clarity. Did the authors mean the live snails were distinguished and collected from the dead ones by the crawling method?

146 Please clarify regarding the pleopod. I don’t think snails have pleopods, which are forked appendages typically found in crustaceans and are used for swimming.

151-152 Please clarify what you mean by ‘except for the reverse primer of primer pair 10 and the primer pair 14, which were not consistent with the literature.” So did you mean you did not design them, or you did not include them in amplifying the specific fragments?

199-208 (1) Except for MP, how did you identify the optimal model for DNA substation for each tree construction method? Did you test for the optimal model, or did you just select a default model? If you use a selected model, did you do a partition homology test for uniform phylogenetic signal? (2) Did you use a single model for the entire genome, or did you partition the genome into segments, with each segment having its own model of DNA substitution?

Reviewer #2: consider current insights for annotation of molluscan mitogenomes as indicated in comments to authors. Specifically, employ visual check after automated mitos annotation, especially considering recommendations as provided by Fourdrilis et al., 2018; doi: 10.1038/s41598-018-36428-7 and Ghiselli et al 2020; DOI: 10.1098/rstb.2020.0159. Inspection of Genbank entries indicates potential errors in annotation. (That does not impact the phylogenetic analyses). See comment to authors.

**Results**

-Does the analysis presented match the analysis plan?

-Are the results clearly and completely presented?

-Are the figures (Tables, Images) of sufficient quality for clarity?

Reviewer #1: The analysis used was adequate. I have a few comments about the way some of the results are presented.

For Figure 3:

A more effective way to present the phylogenetic trees is to select one tree (e.g. ML tree) and incorporate the bootstrap values of all the tree construction methods for all the clades that appear in all. 

Please remove ‘.1’ in the accession numbers.

Please make sure that the figure can stand on its own. For instance, mention in the caption the outgroup taxon, what the values on node signify, and what the scale bar at the base of the tree mean.

For Table 2:

You need to add a column to indicate what region/gene of the mitochondrial genome each primer pair amplifies and their expected PCR product range in bp.

Reviewer #2: no concerns, see comments to authors

**Conclusions**

-Are the conclusions supported by the data presented?

-Are the limitations of analysis clearly described?

-Do the authors discuss how these data can be helpful to advance our understanding of the topic under study?

-Is public health relevance addressed?

Reviewer #1: Though the results provide the first full mitochondrial genomic sequence for the Yunnan strain of O. hupensis robersoni, the heterogeneity they present for the snails based on the molecular data does not explain how the snail can be controlled. I think the discussion needs to incorporate how the genetic data can actually be used for developing strategies for schistosomiasis monitoring and control.

Reviewer #2: Are the conclusions supported by the data presented?

Data presented do not support statement of noncoding AT region, clarification is needed whether authors refer to collection of small intergenic regions. Otherwise no concerns. 

Are the limitations of analysis clearly described?

no concern although the paper can more clearly contrast the benefits, drawbacks of phylogenetics employing mitogenome versus nuclear genome data

Do the authors discuss how these data can be helpful to advance our understanding of the topic under study?

modestly

Is public health relevance addressed?

Role of O hupensis as vector for schistosomiasis is considered, discussion could include consideration of how genetic diversity relates to disease transmission

**Editorial and Data Presentation Modifications?**

Reviewer #1: The results and discussion sections can be merged to minimize repetitiveness of the data/results mentioned. Other minor editorial modifications are listed below.

Line no. Comment

53 Change to ‘Philippine’

57 Change to ‘sub-branches’

75-77 All subsequent Schistosoma after Schistosoma japonicum can be abbreviated to S.

83 Change to ‘inhabit’

89 Add ‘and’ after ‘(O. h. tangi),’

97 Change to ‘Province’

98 Remove ‘,’ after ‘China’

101 Shouldn’t ‘province’ be capitalized?

106 Change ‘in areas of’ to ‘particularly in areas such as’

124 Remove ‘,’ after ‘O. hupensis’

141 Add ‘in’ after ‘presented’

142 Remove ‘were’

175 Did you mean ‘mitochondrial genome’ for the ‘mitochondrial gene’ mentioned here?

176 Please indicate here the subspecies of the reference O. hupensis mitochondrial genome.

190-198 Please remove ‘.1’ in all the accession numbers.

230 Remove ‘was’

236 Add ‘in’ after ‘presented’

237 Change ‘weas’ to ‘were’

244 Change to ‘is presented in Figure 3.’

249 Change to ‘sub-branches’

255 Add ‘regions’ after ‘southern’

265 Change to ‘Branch’

275 Change to ‘Discussion’

277 Change ‘is’ to ‘are’

279 Add ‘useful’ after ‘might be’

283 Change to ‘results’

290 Add ‘and’ after ‘maternal inheritance’

361 Change to ‘O. hupensis. This…”

434-601 All generic and scientific names in the list of references should be italicized.

Reviewer #2: Revisions invited are textual, (also ask for additional technical details) but also include some re-analysis of annotation. I consider this a major revision.

**Summary and General Comments**

Reviewer #1: The study provides the full mitochondrial genome of O. hupensis robertsoni from Yunnan Province, which allows its placement among the different strains of O. hupensis robertsoni. Though the writers hinted at the possibility of using the genetic information to control the snail, which is an important vector of Schistosoma japonicum, this was not clearly expounded in the paper. I would have wanted that this be further discussed in order to justify obtaining the full mitochondrial genome from 26 individuals.

I have also listed below some minor comments to help improve the paper.

Line no. Comment

110-112 It is not stated why heterogeneity of the snail can pose ‘challenges to monitoring and controlling the snails’. Perhaps the authors need to explain this briefly in order to justify the need to explore genetic variation among the different populations of O. h. robertsoni.

114-118 The authors should mention that the 16S and 12S markers were also used to differentiate the subspecies of O. hupensis. These include the following: 

Chua JC, Fontanilla IKC, Tabios IKB, Tamayo PG, De Chavez ERC, Agatsuma T, Kikuchi M, Kato-Hayashi N, Chigusa Y, Fornillos RJC, Leonardo LR. 2017. Genetic comparison of Oncomelania hupensis quadrasi (Mollendorf, 1895) (Gastropoda: Pomatiopsidae), the intermediate host of Schistosoma japonicum in the Philippines based on 16S ribosomal RNA sequence. Science Diliman 29(2): 32-50.

Okamoto M, Lo CT, Tiu WU, Qui D, Hadidjaja P, Upatham S, Sugiyama H, Taguchi T, Hirai H, Saitoh Y, Habe S, Kawanaka M, Hirata M, Agatsuma T. 2003. Phylogenetic relationships of snails of the genera Oncomelania and 

Tricula inferred from the mitochondrial 12S RNA gene. Japanese Journal of Tropical Medicine and Hygiene. 31(1):5-10.

Wilke T, Davis GM, Dongchuan Q, Spear RC. 2006. Extreme mitochondrial sequence diversity in the intermediate schistosomiasis host Oncomelania hupensis robertsoni: Another case of ancestral polymorphism? Malacologia. 48:143-157.

256-258 Please describe in the text the barriers you mention that separate the two sub-branches. Perhaps you are referring to what you described in your discussion section regarding the Jonsha River Watershed?

318-325 Maybe you can also refer to other previous studies that used other markers such as 16S and 12S genes, notably Chua et al. 2017. They also analyzed several sequences of O. hupensis robertsoni using 16S.

350-351 Please clarify what you mean by ‘unique natural factor in the Jinsha River Watershed or geographic isolation’. Please provide details regarding these and why they could contribute to the genetic differentiation of the two subgroups. You can hypothesize, which can be tested in subsequent research.

359-360 Isn’t the genetic difference big enough to warrant separating them into two subspecies rather than subgroups with O. hupensis robersoni? Do you have threshold values for delineating subspecies based on the whole mitochondrial genome? For the COI gene, some authors suggest using the 3% threshold, though this may vary depending on the taxonomic group.

Reviewer #2: This report describes characterization of full mitogenomes from a number of Oncomelania hupensis robertsoni (Ohr), vector snails for the parasite Schistosoma japonicum, field collected from an endemic area in China. Sequence data were used in phylogenetic analyses relative to previous GenBank entries for Oncomelania hupensis mitogenomes to investigate population structure in the field. Results may inform transmission dynamics, tracking of populations and future control efforts.

Notably, the mitogenome sequencing did not rely on computational assembly of short reads but employed a robust approach involving PCR amplification and Sanger sequencing of overlapping amplicons to generate genome assemblies with high confidence. This helps avoid computational bioinformatics errors that may result with automated assembly of short reads, such as “stacking up” of repetitive regions (e.g. Sharbrough et al., 2023; https://doi.org/10.1093/molbev/msad007) or incorporation of NUMTs (e.g. Wei et al, 2022; https://doi.org/10.1038/s41586-022-05288-7)

The authors are encouraged to include a consideration of how the diversity of these snails and other previously characterized snails reflect the transmission dynamics (efficiency, intensity)of schistosomiasis. 

Some wording and aspects of experiments can be improved for clarity of the paper. Authors are asked to address the following comments:

Abstract,

delete “hilly regions” or use alternative to indicate geographical barriers?

Delete “the” from the genetic differentiation; the mitogenome does not inform on nuclear genome differentiation

Considering that no data resulted from one location, suggest to edit from 14 to “13 villages in Yunnan”, 

Similarly of 30 snails collected, only 2 individuals were used for mitogenome sequencing, adjust numbers accordingly

The non-coding sequence rich in AT is not evident in description, figures or GenBank entries. Provide relevant information.

Mentioning of the outgroup: as “the Phillipines genotypes” is a confusing designation; and seems an unnecessary detail for the abstract. 

Was the comparison/phylogenetic analysis based on full-length genomes?

Instead of homology, a better term mat be identity when comparing DNA sequences. 

Introduction

93 delete “fully”. The scope of this paper cannot provide that.

97,99 Yunnan Provinces or Province, two or one entities? Also considering the map, is this a West-region or South-West region of China.

118 These references deal with blackflies, as possible, cite relevant papers from molluscan work.

119 Include mention of recent genome report (Liu et al 2021; 10.1186/s40249-024-01187-3)

Material and Methods.

Again, consider making numbers consistent for number of sites, snails collected per site, live snail selected by crawling methods, ? number of snail extracted, the 2(?) snails sequenced per site.

140 habit = habitat?

144-5 Genus name in full at start of sentence. Also edit sentence for meaning. Provide brief explanation of “crawling method” additional to citations.

146-7 suggest edit “of a single snail” of individual snails”. How were 2(?) snails selected from the 30 snails/site for mitogenome sequencing? Indicate the specific “Qiagen extraction kit” (the website lists over 30 different kits)

149 edit to “The entire mitogenome was…”

151-2 explain this cryptic statement. Indicate what gene or region was different sequence-wise from “the literature [931]. How was it different and what was the origin of the snail used in that study?

155 identify the type/kit of polymerase used. Importantly, instead of volumes indicate final concentrations of the reagents 

162 explian bimodal mutation as sequencing problem.

164 “to splice” = “to contig assemble”?

170 submission is not for alignment and direction confirmation? Does this refer to identification of open reading frames?

171-174 sentence not informative, delete?

174 “nearest” = “most similar”?

177 if sequences were trimmed, what information of the circular mitogenome was deleted? If no removal, suggest delete the term “trimmed”. Perhaps state, the mitogenomes sequences were oriented with the start of cox1 in the first position.

178 define self-test.

182 Mitos automated sequencing is a good start but annotation requires visual checking: for recommendations see: Fourdrilis et al., 2018; doi: 10.1038/s41598-018-36428-7 and Ghiselli et al 2020; DOI: 10.1098/rstb.2020.0159 . For instance, your Genbank entry OR661787: the region between ND6 and CYTB includes one more potential start codon that would extend the CYTB ORF by three nucleotides and one AA,. Ignoring this start codon goes against the genetic code, and reports a trunctated CYTB 5’ sequence, and provides an incorrect start codon (ATG versus ATT). This also impacts the discussion of the number of nucleotides in intergenic, non-coding regions. Please check ALL gene ORFs and revise accordingly.

ND6 CYTB

 I * (M) M R

ATT TAG C ATT ATG CGC

2.5 Construction of phylogenetic tree: what alignment program was used, were the mitogenomes aligned full length or was the analysis based on a partial sequences?

NJ is not frequently used, what is the rational for including it here?

2.6 Sequence genetic distance and homology analysis.

Again indicate if analyses used alignment of full length versus partial mitogenomes.

Results

227 26 = 2 for each collection site? Explain the problem to obtain the/any sequences form Dianzhong

232 delete “significantly”. Not tested statistically and invertebrate mitogenomes typically have A+T >G+C.

234 “a non-coding region rich in A+T”. It is unclear from the paper if this is a region independent from the intergenic regions described in 237-239.

If this a region such as described by Sharbough et al., 2023; https://doi.org/10.1093/molbev/msad007, provide details in the paper for location, length, possible repeat content. If this “region” refers to the intergenic regions described in 237-239, correct/revise the manuscript accordingly to clarify this unequivocally.

Include description: were the two mitogenomes from each location most similar to each other or do comparisons reveal another more complex pattern? Can this be indicated in the (legend of the) tree? 

3.2 Phylogenetic tree construction:

Provide the other 3 trees as supplementary results

Suggest to indicate these groupings on the map in figure 4.

268: homology or identity?

271-3 reword sentence to clarify “formed a rectangle in the lower right corner”.

Discussion

280 edit to “collected 14 representative, and recovered sequences from snails from 13 of these locations”. Provide interpretation for why these sequences were unsuccessful?

286-291. Include the contrasting concept that mitogenomes also have drawbacks and may be used complementary with nuclear genes for optimal results.

292-297 Information seems duplicated from introduction. Reduce in one of these sections.

323 what is “foregoing”?

326 edit “could be subdivided” to “is subdivided”e 

341 avoid contractions

346-357, provide hypothesis/es for lack of diverhgence despite obvious geographical barriers; can there be transport of snails by wildlife or human activity? What is the age of the barriers…? Other?

366-369 OK to list the components of the effort but it seems unusual to state that these are major strengths of this study, Suggest rewording. Suggest mentioning that the methods (PCR, sequencing of overlapping sequence regions for assembly) avoid the computational challenges and possible errors of computational assembly of short reads, see Sharbrough 2023).

370- clarify how elimination of some snail is a limitation to this study. What hypothesis do you have of how that mitogenomic information would have shaped understanding of phylogeny?

383 edit “These finding may provide the..”: to “These findings provide ..” 

388 edit “used to the important” to “as”

PLOS authors have the option to publish the peer review history of their article (what does this mean?). If published, this will include your full peer review and any attached files.

Reviewer #1: No

Reviewer #2: No
---

## [Decision Letter · Decision Letter 1]

3 Oct 2024

Dear MD Dong,

Thank you very much for submitting your manuscript "Genetic differentiation of Oncomelania hupensisrobertsoni in hilly regions of China : using the complete mitochondrial genome" for consideration at PLOS Neglected Tropical Diseases. As with all papers reviewed by the journal, your manuscript was reviewed by members of the editorial board and by several independent reviewers. The reviewers appreciated the attention to an important topic. Based on the reviews, we are likely to accept this manuscript for publication, providing that you modify the manuscript according to the review recommendations. 

Sincerely,

Luc E. Coffeng, MD PhD

Academic Editor

Audrey Lenhart

Section Editor

Reviewer's Responses to Questions

**Key Review Criteria Required for Acceptance?**

**Methods**

-Are the objectives of the study clearly articulated with a clear testable hypothesis stated?

-Is the study design appropriate to address the stated objectives?

-Is the population clearly described and appropriate for the hypothesis being tested?

-Is the sample size sufficient to ensure adequate power to address the hypothesis being tested?

-Were correct statistical analysis used to support conclusions?

-Are there concerns about ethical or regulatory requirements being met?

Reviewer #1: Lines 157-163: Change to “Dead O. hupensis snails were selected and removed using the crawling method. Briefly, the snails were placed in the center of a petri dish with a diameter of 15cm, after which a few drops of water were poured over the snails. A mesh cover was placed over the petri dish to prevent the live snails from crawling out. The set up was placed ar room temperature (25℃) over 24 hours, after which the ones that crawled from the center were regarded as live. Thirty live snails were used for DNA extraction.

Line 164: Please indicate the approximate size of the head-foot muscle used for DNA extraction (i.e., 2-mm strip, etc.)

Lines 212-221: based on the reply of the authors, they must include the model of DNA substitution used for ML and NJ. For example, they used the GTR+G model for ML. They must also specify the default models they used for NJ.

Reviewer #2: (No Response)

**Results**

-Does the analysis presented match the analysis plan?

-Are the results clearly and completely presented?

-Are the figures (Tables, Images) of sufficient quality for clarity?

Reviewer #1: For figure 3, there were some singular values (100) on certain nodes instead of four (for ML/MP/ME/NJ). What do these singular values mean? Are they bootstrap supports for just one tree method? This needs to be clarified. 

Line 343: Remove the extra ‘,’ after ‘Additionally’

Reviewer #2: (No Response)

**Conclusions**

-Are the conclusions supported by the data presented?

-Are the limitations of analysis clearly described?

-Do the authors discuss how these data can be helpful to advance our understanding of the topic under study?

-Is public health relevance addressed?

Reviewer #1: For the conclusion section: The response of the authors regarding how the mitochondrial heterogeneity of O. hupensis robertsoni from the samples they collected be correlated to their control was sufficient; however, it was not detailed much in the main manuscript. I hope the authors can better incorporate their response into the main manuscript

Reviewer #2: (No Response)

**Editorial and Data Presentation Modifications?**

Reviewer #1: None.

Reviewer #2: (No Response)

**Summary and General Comments**

Reviewer #1: I have no further comments. The other technical comments have already been raised by the other reviewer, and I will leave it to the other reviewer to determine if the authors have sufficiently addressed the comments.

However, I would like to the authors to carefully consider their conclusion regarding the applicability of their data for control of the snail. They need to elaborate on this further. Please see my comments on their conclusion section.

I also hope the authors are able to addressed my issue on Figure 3, especially on the singular values on certain nodes as these can be quite confusing.

Reviewer #2: the author response to previous review comments are mostly effective, 

However, the response to comment 27 (included below) remains an issue:

The authors recognize, yet ignore an in-frame start codon upstream of the indicated reading frame for CYTB.

The argument that a previous Genbank uses the same shorter interval is not adequate, unless that annotation is based on protein-level confirmation of CYTB. The previous entry may be incorrect, resulting from not recognizing the 5' alternative startcodon. It is not benefiting science to continue annotation that ignores the "rules" of the genetic code. I repeat the comment that the annotation should interpret the actual genetic code provided by the experimental data, not perpetuate potential errors in previous GenBank entries. 

"Comment 27: 182 Mitos automated sequencing is a good start but annotation requires

visual checking: for recommendations see: Fourdrilis et al., 2018; doi:

10.1038/s41598-018-36428-7 and Ghiselli et al 2020; DOI: 10.1098/rstb.2020.0159.

For instance, your Genbank entry OR661787: the region between ND6 and CYTB

includes one more potential start codon that would extend the CYTB ORF by three

nucleotides and one AA,. Ignoring this start codon goes against the genetic code, and

reports a trunctated CYTB 5’ sequence, and provides an incorrect start codon (ATG

versus ATT). This also impacts the discussion of the number of nucleotides in

intergenic, non-coding regions. Please check ALL gene ORFs and revise accordingly.

ND6 CYTB

I * (M) M R

ATT TAG C ATT ATG CGC

Response 27: Thank you for your kind suggestion. As suggested, we have visual

checked all mitochondrial genome sequences. The base of OR661787 sequence is

GCATT between gene ND6 and CYTB gene. According to the literature provided by

the reviewer, the literature does mention “ATY” as a possible codon. However, when

we aligned the mitochondrial sequences in NCBI, we found that the start codon of the

mitochondrial CYTB gene was ATG, so we used ATG as the start codon as well." 

""

PLOS authors have the option to publish the peer review history of their article (what does this mean?). If published, this will include your full peer review and any attached files.

Reviewer #1: No

Reviewer #2: No

Figure Files:

Data Requirements:

Reproducibility:

References

---

## [Editor Report · Decision Letter 2]

9 Nov 2024

Dear MD Dong,

We are pleased to inform you that your manuscript 'Genetic differentiation of Oncomelania hupensis robertsoni  in hilly regions of China: using the complete  mitochondrial genome' has been provisionally accepted for publication in PLOS Neglected Tropical Diseases.

Best regards,

Luc E. Coffeng, MD PhD

Academic Editor

Audrey Lenhart

Section Editor

Shaden Kamhawi

co-Editor-in-Chief

Paul Brindley

co-Editor-in-Chief

At the proofing stage, please carefully check the wording of your latest revisions. It seems that some of them have introduced new language errors, like in lines 342-343: "These findings may provide important information for formulating targeted of schistosomiasis control measures."

---

## [Editor Report · Acceptance letter]

20 Nov 2024

Dear MD Dong,

We are delighted to inform you that your manuscript, " Genetic differentiation of Oncomelania hupensis robertsoni  in hilly regions of China: using the complete  mitochondrial genome," has been formally accepted for publication in PLOS Neglected Tropical Diseases.

Best regards,

Shaden Kamhawi

co-Editor-in-Chief

Paul Brindley

co-Editor-in-Chief
